# “Shedding Light on Light”: A Review on the Effects on Mental Health of Exposure to Optical Radiation

**DOI:** 10.3390/ijerph18041670

**Published:** 2021-02-09

**Authors:** Davide Elia Bertani, Antonella Maria Pia De Novellis, Riccardo Farina, Emanuela Latella, Matteo Meloni, Carmela Scala, Laura Valeo, Gian Maria Galeazzi, Silvia Ferrari

**Affiliations:** Department of Biomedical & Metabolic Sciences and Neuroscience, School of Specialization in Psychiatry, University of Modena & Reggio Emilia, 41124 Modena, Italy; dbertani92@gmail.com (D.E.B.); amp.denovellis@gmail.com (A.M.P.D.N.); riccardo.farina10@hotmail.it (R.F.); emy.latella94@gmail.com (E.L.); matte_meloni@yahoo.it (M.M.); carmelascala27@gmail.com (C.S.); lally.v@hotmail.it (L.V.); gianmaria.galeazzi@unimore.it (G.M.G.)

**Keywords:** light, light exposure, melatonin, calcitriol, bio-psycho-social, mental health

## Abstract

In relation to human health and functioning, light, or more specifically optical radiation, plays many roles, beyond allowing vision. These may be summarized as: regulation of circadian rhythms; consequences of direct exposure to the skin; and more indirect effects on well-being and functioning, also related to lifestyle and contact with natural and urban environments. Impact on mental health is relevant for any of these specifications and supports a clinical use of this knowledge for the treatment of psychiatric conditions, such as depression or anxiety, somatic symptom disorder, and others, with reference to light therapy in particular. The scope of this narrative review is to provide a summary of recent findings and evidence on the regulating functions of light on human beings’ biology, with a specific focus on mental health, its prevention and care.

## 1. Introduction

Light, a necessary condition for human vision, is a fascinating and complex phenomenon, and is still the subject of study and research. Light is a source of life, of well-being, of energy and positive power, essential to almost everything happening on the earth.

According to the International Lighting Vocabulary (ILV), light is “any radiation capable of causing a visual sensation directly”, while the Illuminating Engineering Society (IES) defines light as “radiant energy that is capable of exciting the retina and producing a visual sensation in humans” (https://www.ies.org/definitions/light/). One of light’s most curious properties is its dual nature as wave and particle, energy and mass [1]. In fact, quantum physics identifies the photon as the constitutive unit of the light, a particle element without mass and electric charge; however, the constant oscillatory motion with which the photon propagates in space assumes the characteristics of an electromagnetic wave, which goes to constitute optical radiations [2]. The electromagnetic radiations are distinguished according to wavelength and frequency, which are inversely proportional to each other; those perceptible by the human eye are set between 380 and 780 nanometers in length and 790 and 435 THz in frequency and constitute the visible spectrum. Other optical radiations include UltraViolet (UV, 100–400 nm) and InfraRed (780 nm–1 mm) radiation.

In nature, optical radiation emission may happen as incandescence or luminescence. Incandescence, typical of celestial bodies (including the sun), fire, and nuclear energy, is the emission of light from a body as a consequence of its high temperature; luminescence, which is also available to some living organisms (such as algae, fireflies and jellyfish) is not associated with heat emission [3]. Optics is the branch of physics studying the properties and features of optical radiation, and includes phenomena such as reflection, refraction, diffraction, and so on. When optical radiations hit a material body, they behave differently, for example being refracted or absorbed in different proportions, depending on the characteristics of the object affected and the nature of the radiation; this mechanism is the basis of the visual perception of the human being, as well as of other forms of interaction between optical radiation and the human body (e.g., through the skin).

Human vision is the ability to extract information about the surrounding space by capturing light radiation through highly specialized cells, the photoreceptors in the retina. Photoreceptors absorb electromagnetic radiation within the visible spectrum, determining the photoisomerization of the vitamin A-derived visual chromophore and its switch from its cis- to trans- configuration [4]. This activates the opsins, light-sensitive proteins that spark a signal transduction cascade leading to sodium channels closure and cell hyperpolarization. Thus, retinal nerves are triggered, and light stimulus is led to the upper levels of the visual system [5]. There are two types of photoreceptors: the cones, located in greater density in the central portion of the retina, are activated in higher environmental light conditions (daylight and electric light), allow for sharp and detailed vision, and are divided into three main categories, each of which is sensitive to a different wavelength of light radiation, which determines the color perception (photopic vision); the rods, located in the retinal periphery, react at low environmental light, do not mediate color perception, and provide night vision [6]. Thanks to photoreceptors, the light stimulus is then turned into an electrical signal and led to the lateral geniculated nucleus of the thalamus, which amplifies the information coming from the retina. There, it is integrated with signals coming from the cortex and the thalamus itself [7,8], modulating phenomena such as visual attention and space-time integration of images. From the geniculate nucleus, the information is further carried to the primary visual cortex, located in the posterior pole of the occipital lobe of the brain, where conscious visual perception takes place [9,10].

Noteworthy is also the anthropological and cultural significance of light, historically linked with positive and beneficial concepts as opposed to darkness, which is instead associated with negative symbolic meanings, such as evil, ignorance, fear and hatred. Humanity’s literary production is rich in examples of this kind: basically, every main religion describes light as a divine and salvific attribute, the highest symbol of the manifestation of God—or even God himself (e.g., in Manicheism). In philosophy, light is the symbol of reason, intelligence and truth, capable of saving mankind from the darkness of ignorance (e.g., the myth of Plato’s cave). In the art field, it is impossible to count how many painters, sculptors, architects and graphic artists have studied, exploited, recreated and worked with light, placing it at the center of their artistic and cultural production.

In this work, the impact of optical radiation (mostly light, but also UV) on human health is addressed, with a more specific focus on mental health, in the form of a non-systematic narrative review. The three main sections constituting the structure of the review provide specific analysis of: effects on circadian rhythms; effects on health via direct exposure to the skin; and effects on humans’ well-being, functioning and mental health. The last section, additionally, includes hints for the clinical use of optical radiation.

## 2. Effects on Circadian Rhythms

### 2.1. Retinal Ganglion Cells, Clock Genes, Melatonin and Serotonin and Their Relationship with Light

Periodic oscillations are typical of biological processes, and some of them are synchronized with the light–dark cycle, usually known as biological rhythms or biorhythms [11]. Every organism has physiological activities that exhibit a circadian rhythm behavior, e.g., the sleep–wake cycle, the circadian production and secretion of cortisol, circadian regulation of body temperature, and many others. The circadian rhythms can be influenced by different external or internal factors, and light, natural or from an artificial source, is one of the most important [12]. Ambient light can contribute to regulating “non-visual” response, through a recently discovered photoreception system made by the intrinsically photosensitive Retinal Ganglion Cells (ipRGCs). They contain the photopigment melanopsin and they receive and integrate signals from the rods and cones [13]. These ganglion cells propagate the signal to the circadian clock in the SupraChiasmatic Nucleus (SCN), located in the hypothalamus, through the retino-hypothalamic tract. The SCN represents the main circadian clock, though many different tissues, including liver, lungs and isolated cells, can express periodic changes of function without its control [14]. Interestingly, ipRGCs can be strongly activated by light with wavelengths of ~480 nm, such as the blue spectrum of light [15,16], while different wavelengths, such as green light, are capable of the activation of photopigments [17].

Light exposure during night-time can induce phase-shifts, through the induction in the SCN of some clock genes and the circadian cycle derives from a complex feedback loop interaction between clock genes and their proteins [18]. In mammals, the SCN can be synchronized not only with light, but also with other non-photic factors, such as serotonergic pathways activation [19], thanks to many serotonergic receptors expressed in the SCN [20]. Any alterations in serotonin input to the SCN may change the circadian rhythms, nevertheless it is still unknown how the disruption of the entire system can exactly affect the SCN and clock output [21]. In addition to serotonin, many other neurotransmission systems show rhythmic variations, regulated by the SCN, such as the noradrenaline, histamine, GABA and acetylcholine ones [22].

Of interest, recently it has been hypothesized that light can directly influence emotional brain processes, through direct projections from ipRGC, and classical retinal ganglion cells, to many brain areas, such as the hypothalamus, thalamus and amygdala. In turn, there are many other projections to subcortical and cortical areas involved in behavioral and physiological functions [17,23,24].

Finally, the SCN controls the timing of release of the melatonin (MLT) hormone. MLT suppression represents a marker of circadian light sensitivity. The relationship between light and MLT seems to be related with serotonergic pathways: the administration of the Selective Serotonin Reuptake Inhibitor (SSRI) antidepressant citalopram, which does not affect baseline MLT levels, was found to increase MLT suppression and was associated with a delayed MLT onset in normal lighting conditions. This suggested that SSRIs may increase the human circadian system sensitivity to light, not through ipRGC activation, but through serotonin effects in the SCN [25,26].

Seasonal Affective Disorder (SAD) is mentioned in the 5th version of the Diagnostic and Statistical Manual of Mental Disorders (DSM) as a specifier of recurrent depression or bipolar disorder, whether episodes of major depression have a seasonal pattern. Both altered monoaminergic transmission and a “chrono-biological hypothesis” linking development of SAD to exposure to light and MLT levels have been discussed as neurobiological pathways to SAD [27]. Many SAD patients seem to be phase-delayed in their chrono-biological cycle and data such as abnormal diurnal MLT levels, prolonged nocturnal MLT production in the winter season and delayed release of this hormone have also been documented in these subjects [27].

Light pollution is defined as the presence of artificial (or anthropogenic) light in an environment that would otherwise be dark, usually the night outdoors. It is a direct effect of urbanization and has been criticized for its many detrimental effects on biology, aesthetics and health. Sometimes referred to with the acronym ALAN (Artificial Light At Night), it was proved to be responsible for circadian disruption of MLT synthesis, through the SCN and the peripheral oscillators [28,29,30], so light may represent a risk factor for developing metabolic, endocrine and/or oncological diseases. Light pollution due to ALAN and rotating shift work have been studied for their role on circadian disruption, which seems to be related to serious health problems consequently to the dysregulation in leptin [31], ghrelin [32], insulin [33] and glucagon [34] levels and patterns of secretion. Several human and non-human studies have shown a modulated metabolism at the molecular, physiological and behavioral levels by the SCN [35,36]. In a human study in patients with chronic nonalcoholic liver disease, after MLT treatment, they showed significant increase in plasma levels of ghrelin, adiponectin and leptin [37]. In human brown and white adipose tissues, MLT receptors are expressed, so they may have a regulatory metabolic role. ALAN-induced suppression of MLT may be associated with obesity, through brown adipose tissue reduced metabolic activity [38,39], which is important for body mass regulation [40,41], whereas MLT increases brown adipose tissue activity, decreasing body mass [42].

According to the following findings, dysregulation in clock genes expression seems to also be related to tumorigenesis processes: in vitro, knocking down clock genes causes an increased expression of a network of cancer-related genes and decreases the activity of several tumor suppressor genes related to breast cancer [43]. In another in-vitro study, upregulation of other clock gene transcription blocked cell proliferation in breast cancer [44] via degradation of estrogen-receptor-α [45]. Several studies have demonstrated a causal link between circadian disruption and breast cancer risk: increasing the number of years of shift work is associated with increased risk of cancer [46]. The higher ALAN intensity is, the higher the risk of developing breast cancer is [47]. MLT also has anti-apoptotic and anti-carcinogenic effects [48,49,50,51].

### 2.2. Light Effect on the Endocrine System

Analyzing the impact on the endocrine system, the main hormone involved is MLT. MLT has two types of secretion: one is discontinuous, responding to the circadian clock stimulation, by the SCN, in the pineal gland, in dark condition. The other is continuous, by the cutaneous melatoninergic system [52]. MLT onset is the starting point of the biological night, regulating the sleep–wake cycle. Secondary feedback generates reciprocated signals between the circadian clock and metabolism, so MLT bioactivity is involved in many other behavioral, endocrine and immune processes [53,54].

Adrenal gland function is well-known for being regulated by the circadian system, with higher glucocorticoid levels in the morning, right before awakening, and lower during the day. MLT respectively increases and decreases progesterone and estradiol levels, whereas ALAN has been associated with increased secretion of estradiol. This is another biological pathway through which light may have an impact on affective regulation, considering for example that glucocorticoid dysregulation has been associated with mood disorders (more specifically, hypercortisolemia and reduced reactivity of the negative feedback system in patients with major depression) or the role of neurosteroids derived from progesterone (allopregnanolone, allotetrahydrodeoxycorticosterone) on mood disorders [55,56]. Derivatives of progesterone have demonstrated significant mood-stabilizing effects: pregnanolone was found to be effective and safe in the treatment of bipolar depression [57]. Indirectly, it has been hypothesized that light pollution may play a role in the genesis of bipolar disorder, by decreasing MLT production, thus leading to dysregulating levels of steroid hormones [58].

Finally, humans living in polar regions are at risk of the so called “polar T3 syndrome”, a condition featuring chronically low levels of blood T3 and leading to psychological disorders, such as depression and increased aggression, similar to SAD [59]. Thyroid dysregulation might be caused by circadian disruption, typical of those remote regions.

### 2.3. Other Effects on Health Related to Circadian Rhythms

Recent studies have confirmed that circadian disruption is also involved in the inflammatory pathway leading to metabolic syndrome [60], through alteration of glucose homeostasis, and to obesity [61], but it is unclear if this happens through direct circadian mechanisms, similar to those described above, or more indirectly, e.g., as a consequence of eating behaviors of shift-workers; evidence is still controversial, and very few control studies on workers not involved in shifts exist. In a recent study by Obayashi and colleagues, BMI and serum triglycerides seemed to correlate with exposure to electric night-light [62]. Prasai et al. showed the vicious cycle involving circadian disruption and metabolic syndrome: the first one promoting obesity, which itself causes alteration of sleep quality and even more disrupted metabolic mechanisms [63]. Genetic variation in clock function could influence susceptibility to metabolic effects of such dysregulated pathways, explaining individual and ethnic variations [64].

It is known that causes for circadian disruption may have many consequences on human health, and mental health more specifically, but more studies are needed to better understand how much light exposure is directly involved, and the specific pathophysiological mechanisms; the recent work of Rea and colleagues can help to better understand significant advancements in regard to neurophysiology associated with circadian phototransduction [65]. Many other factors, such as daily patterns of food intake, activity, type and time of light exposure, could be relevant in determining such complex neurobiological pathways. A better understanding of these is extremely wanted for the potential applications in more targeted and etiologic treatments.

## 3. Effects of Light on Health via Direct Exposure to the Skin

Optical radiation plays an important role for the functioning of the skin and, consequently, on complex regulation of body physiology. The disappearance of rickets as a public health issue after understanding the importance of sun exposure is one of the most impressive results of such knowledge. Light, therefore, is an essential regulator of human physiology not only via the circadian rhythms as described previously, but also thanks to biochemical processes starting from epidermis down to dermis and inner teguments, addressed as follows.

Vitamin D, or calcitriol, is a fat-soluble vitamin that is synthesized up to 90% in the skin thanks to the exposure to sunlight, more specifically to UVB light stimulation [66]. Moreover, in the epidermal layers, synthesis of catecholamines and acetylcholine is observed. In a review by Holick [67], human keratinocytes, after UV radiation exposition, showed a marked increase in the expression and production of beta-endorphin, so exposure to optic radiation (via UV skin stimulation) might in fact contribute to increase feelings of wellbeing and pain relief and, on the contrary, sunlight deprivation could favor depression [67].

A 2010 study described indispensable enzymes and receptors located in different parts of the human brain that are regulated by Vitamin D [68,69]. For example, calcitriol activates the gene expression of the enzyme tyrosine hydroxylase, the rate-limiting step in the synthesis of the catecholamines, resulting in an increase in the amount of circulating dopamine, noradrenaline and adrenaline. Calcitriol may also play a role in the functionality of cholinergic enzymes by increasing the activity of choline acetyltransferase (needed for acetylcholine synthesis) and removing the block of acetylcholine synapse transmission operated by acetylcholinesterase. The role of dopamine, noradrenaline and acetylcholine in the pathophysiology of mood disorders is well established [70,71,72]. New studies are focused on further roles played by Vitamin D within the central nervous system: for example, on how calcitriol increases availability of some neurotrophins, like nerve growth factor. This could be relevant to explaining the pathophysiology of schizophrenia, which may be accompanied by a defect in the development of brain tissue due to failing functioning of neurotrophins [73]. Depression and fatigue can occur as long as a deficiency of Vitamin D persists in otherwise healthy individuals, a mechanism that may be relevant in SAD.

As mentioned previously, MLT is not only rhythmically secreted by the brain, but also continuously by the cutaneous serotoninergic/melatoninergic system. The enzyme arylalkylamine N-acetyltransferase, allowing serotonin acetylation, generates N-acetylserotonin. This one is then converted into MLT, through methylation by the enzyme hydroxyindoleO-methyltransferase [52].

## 4. Effects of Light on Well-Being, Functioning and Mental Health

Life-style and environment play a significant role in the relation between light and human health: relevant examples include the epidemiology of rickets mentioned above, but also, since the 1980s, the health alert about potential dangers connected to excessive sun exposure and risk of skin cancer (in particular malignant melanoma).

Since the second half of the 20th century, in western developed countries and among populations living in temperate climate regions, indoor activities have become largely prevalent in all age groups and for the greater part of the year, with sun exposure limited to holiday time in the summer. In parallel with this change of UV exposure habits, an increasing prevalence of major depression in the US and in Europe has been reported.

Built environment affects mental health both directly and indirectly [74]. Features of the built environment include private housing (house types, floor levels, house quality, neighborhood quality), institutional and public settings (including health care facilities), density of population/crowding, noise, and indoor air quality, but also, relevant to this review, lightning. Increasing attention has been paid recently to how different features of the environment, including human architecture, can influence individuals’ mental health and, therefore, to effective ways to change the environment in order to potentiate beneficial effects of light on health. Levels of illumination, for example, the amount of daylight exposure allowed by windows or balconies according to their dimensions, as well as to the closeness of buildings, impact massively on psychological well-being. Individuals chronically exposed to shorter hours of daylight in built environments tend to suffer more from sadness, fatigue, and clinical depression. Patients hospitalized for severe depression recover more quickly in sunny versus poorly lit rooms. Levels of illumination, and not spectral frequency, appear to be the critical element in SAD [75]. Insufficient exposure to daylight from windows disrupted normal circadian rhythms of cortisol in Swedish school children [76]. Distractibility and cooperative social behavior in the classroom were also adversely affected. Though it may be argued that changing architecture is not relevant to light, by itself, architecture could be altered within the space and become a lighting intervention. Suggestions may include the position of desks (e.g., in offices or schools) or beds (e.g., in hospitals or nursing homes) according to the windows to receive more light or with a better angulation, as well as the choice of colors for the wall paintings considering reflection of light in a certain indoor environment.

Outdoor activity is known for its potential benefits to mental health and subjective well-being [77], though it may be difficult to understand the specific contributions of being exposed to a natural environment or to natural light, since the two are combined [78,79]. A 2016 study tried to understand whether there was a correlation between exposure to sunlight and the mental health of workers. A distinction was made between direct and indirect light as they would have different actions on the human body, for example only direct light would be correlated with vitamin D-related effects [80]. Exposure to direct sunlight, in the sense of outdoor exposure, was correlated with lower levels of anxiety, while exposure to indirect sunlight, through windows, was correlated with lower levels of depressed mood [81]. Other studies have tried to identify what specific elements of a natural environment might be significant in explaining such modulating role, by means of methods to reduce reciprocal confounding effects (e.g., by showing only images of landscapes, rather than direct exposure, which necessarily also implied the presence of sunlight) [81,82].

Stress, mood and mental health in general were the clinical targets most commonly assessed.

As to stress, neurofunctional correlates were found suggesting that the hippocampus and the amygdala areas, which are connected to the increase in working memory and stressful emotions, are activated when viewing urban environments while the vision of rural environments activates the basal ganglia, more associated with the experience of pleasure [83,84].

As to mood, many studies have confirmed the positive role of exposure to a natural environment and to natural light [78,79,81,82]. Nevertheless, this beneficial effect was more evident when comparing exposure to a natural environment and to a markedly urban one, rather than when comparing exposure to a natural environment and a more neutral one [82].

With regard to mental health in general, this was studied in terms of incidence of mental disorders, work productivity, and clinical and psychological well-being [78,79,81,82]; mood and anxiety disorders were found to be more common among urban dwellers and the incidence of schizophrenia was much higher in people born and raised in cities, even more if in degraded contexts and overcrowded housing [74,79,82,83,85].

Despite the great heterogeneity in research methods, almost all studies agree in recognizing a positive influence of architectural aspects of the environment on mental health, for example, windows guaranteeing adequate exposure to open air and sunlight, which emerged as the single most important feature that could make a difference [79,84].

Among the components of an urban environment that could affect mental well-being negatively, the following were included: multi-family dwelling, living in upper floors, poor quality of dwelling and neighborhood, poor control, inadequate social support [74]. While adequate exposure to rural environments may be not feasible in vast urbanized contexts of living, working on some of these elements could still prove possible to positively modulate the mental status of inhabitants, or workers, or patients in hospital facilities. In some cases, technology may offer support, providing effective artificial reproduction of natural light, such as Solid-State Lighting (SSL) [78,79,86]. Another example is an automatized system of regulation of intensity of light, colors and music of a certain living place based on an “Emotion Detection” system, an A.I. tool sensitive to the facial expressions and behaviors of human beings [87]; futuristic as these systems may sound, they could prove to be valid support in the promotion of well-being. A recent study also highlighted that intense and excessive home lighting, especially in the evening hours, would have a significant impact on sleep quality and, consequently, on the well-being of individuals [88].

Light, with particular regard to daylight, is a fascinating field that has not yet been fully explored. Several disciplines and professions could benefit from an improvement in the knowledge and understanding of how light interacts with human beings, both physiologically and psychologically [89].

### Light as Therapy for Mental Disorders

Considering the many relationships between light exposure and health-relevant regulatory mechanisms in the body and nervous system described above, a natural consequent assumption may be that light may serve as a vehicle of therapy.

Light therapy, also called bright light therapy, or phototherapy, is a treatment that provides therapeutic effects through exposure to electric light that simulates sunlight. Early studies used bright white light, similar to daylight, to analyze light effects on human circadian rhythm. Subsequently, it was demonstrated that short wavelength blue light has more major effects on phase shifting than the rest of the visible light spectrum [90]. The use of low-intensity light has been suggested to produce more antidepressant efficacy and fewer compliance issues [91].

At present, the standard method consists in exposing the subjects to a “light box” placed at eye level and at a distance of about 30–80 cm (according to the recommendations of the device); the onset of beneficial effects usually occurs about a week after the start of treatment [92]. The treatment is usually carried out in the morning, but several studies have evaluated the possibility of evening treatment schemes based on the patient’s chronotype. Some studies have evaluated how, in bipolar disorder, midday light exposure generates a reduced risk of manic switch compared with morning exposure [93,94]. Terman et al. determined that the best time to administer light therapy and implement its antidepressant effect is 7.5 to 9.5 h (average of 8.5 h) after the onset of MLT in plasma. The time of MLT onset can be inferred by means of the Morningness–Eveningness Questionnaire (MEQ), which has been validated to reflect circadian rhythm phase advance [95,96].

For SAD and non-seasonal unipolar depression, cycles of 30 min/day with intensity of 10,000 lux are set; in case of administration of lower intensity light, the daily duration must be increased (for example, 5000 lux for 1 h/day or 2500 lux for 2 h/day) [92]. For non-seasonal bipolar disorders, cycles at 5000 lux are generally arranged; the duration is progressively increased by 15 min each week, until the target of 1 h/day is achieved in about a month, also taking into account effectiveness of the treatment and patient tolerance [97].

In the past few years, new lamp configurations in the form of visors have been developed to allow greater patient comfort. One study has shown promising effects on circadian phase shift [98]; however, other studies have shown no benefit over dim light exposure [99,100,101]. Further studies will be needed to investigate the actual clinical applications of these visors.

Light therapy can also be self-administered at home, through therapeutic schemes programmed by the doctor and closely monitored according to clinical changes, to optimize the therapeutic scheme (timing and dosages) [96]. Light therapy was also proposed and studied as an add-on to antidepressant therapy, especially for drug-resistant forms; in these cases, it is important to evaluate possible changes in the dosage of the drug, and if the patient takes a photosensitizing drug, it may be necessary to reduce both the dose of the drug and the light [96]. Better results were obtained by administering light in the morning or in association with sleep deprivation. Additionally, it was seen that light therapy may start improving mood earlier than antidepressant drugs [102].

The need for maintenance treatment regimens differs according to patient response: some patients maintain complete remission of symptoms after a single course of therapy [103], others experience symptomatic relapse after a variable period of 1–3 weeks, and some prefer to continue treatment throughout the winter period [104].

The side effects of this treatment are usually mild and reversible: headache, eyestrain, blurred vision, photophobia, irritability, diarrhea and nausea are sometimes reported. If therapy is administered in the evening, initial insomnia and hyperarousal may occur. Side effects may be in part associated with certain parameters of light exposure, such as intensity and duration of exposure, timing, spectrum, method of exposure, and angle of incidence of the light spot relative to the eye [96]. Several cases of development of manic/hypomanic episodes during the treatment have been reported in the literature: Tuunainen et al. found that hypomania was the only side effect among patients receiving light therapy versus controls [102]. To reduce the risk of manic switch, combination therapy with mood stabilizers is indicated in bipolar patients [105]. Very rarely, other side effects have been reported: hot flashes [106,107], menometrorrhagia [108], transient and mild uterine bleeding [96], and flare-up of previous trigeminal neuralgic syndrome [109]. Side effects can be limited through the decrease of the light dose (intensity and/or duration), the increase of the distance between the patient and the light source, or the reduction of the timing of the sessions [110].

As to the effectiveness of light therapy, after the first use in SAD by Rosenthal in 1984 [75], several studies were conducted in the following years in which an attempt was made to further understand the effects and possible targets of this treatment. The main outcomes of these studies are summarized in Appendix A [111,112,113,114,115,116,117,118,119,120,121,122,123,124,125,126,127,128,129,130]. Some studies [131,132,133] have also shown that light therapy is more likely to have beneficial effects on forms of SAD with atypical symptoms. A recent study by the Cochrane [134] evaluated the preventive effect of light therapy on the onset of depressive symptoms of SAD when administered a few weeks before the usual relapse period.

Even if light treatment is reported as the treatment of choice for SAD, it may as well have a positive impact for non-seasonal forms of depression and other psychiatric conditions including bipolar disorder [135], depression in elderly subjects [136,137,138], depression in Parkinson’s disease [139], ante-/postpartum depression [140], and depression in adolescent patients [141]. A selection of the most relevant studies referring to this is offered in Appendix A [97,105,135,142,143,144,145,146,147,148,149,150,151,152,153,154,155,156,157,158,159,160,161,162].

Children and pregnant women could be select targets of light therapy, since use of psychotropic medications is controversial, limited or forbidden in these categories [163]. Such evidence is still limited and at a preliminary stage, which is why the above articles are not included in Appendix A.

Other fields of clinical application of light therapy include: sleep–wake cycle disorders, e.g., nocturnal alertness in night shift workers [90]; premenstrual depression [164]; eating disorders [163]; behavioral disturbances related to dementia [90]; and adult Attention-Deficit/Hyperactivity Disorder (ADHD) [163].

The currently available evidence derives from methodologically heterogeneous studies with relatively small sample sizes [165]. It is believed that further clinical studies of greater quality and with larger sample sizes are needed. Particularly, criticism has been raised on the reliability of data demonstrating superiority against placebo, due to poor methodology [166]. Subsequent attempts to minimize the effect of placebo seemed to confirm the effectiveness of light therapy for SAD [121].

## 5. Limitations

While collecting, organizing and presenting the material of this narrative review, a few limitations need to be acknowledged: first of all, we are aware that the topic is relevant in many different fields of human knowledge, such as physics and architecture, and in many different medical specialties, such as dermatology, oncology, and immunology. While imprecisions and excessive generalizations may be present when touching topics that are away from the authors’ area of expertise, the aim of this review was to provide a general idea of the interconnections, with a specific focus on mental health and potential suggestions for use in the clinical practice.

Moreover, to structure the summary tables, some of the most recent meta-analyses in the literature were identified, from which the articles contained therein were extrapolated; since this is a non-systematic review, no standardized method of selection of these articles was used, but an attempt was made to rely as much as possible on the relevance to the topic. Therefore, some relevant studies in the literature may be missing in these tables.

## 6. Conclusions

Light is a potent regulator of biological functions that activates many complex biological pathways in the human body. Therefore, it may also have significant impacts in mental functioning. Further studies are needed to understand the complex neuro-biology implied, and interconnecting many different neuro-psycho-endocrine systems.

Clinicians can already include light therapy among their therapeutic options, as well as inform and support their patients on relevant actions resulting in more adequate light exposure, mostly outdoor activities, combining exposure to sunlight, open air and physical exercise. A more careful application of these principles to architecture and urban design, as well, may contribute to improved quality of life and health prevention.

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
