# Peer review of "“Shedding Light on Light”: A Review on the Effects on Mental Health of Exposure to Optical Radiation"

_ijerph, 2021, doi:10.3390/ijerph18041670_

Round 1
Reviewer 1 Report
Review for ijerph-1028407
The authors present a narrative review on the effects of light on human health from a “PNEI” perspective. This is a topic worthy of discussion, and the authors bring a unique angle to the field. There were some issues with expression throughout the manuscript, and I have concerns about the current structure of the review and how it minimises/brushes over some points which are particularly important.
I think the paper requires a significant restructure. As it is there are a myriad of sections, many of which are very brief and underdeveloped. The current structure also does not clearly state the primary ways by which light may impact health, these are explored peripherally via a number of underdeveloped sections. I would suggest three major sections: 1. Effects of light on circadian rhythms and subsequent impacts on health, 2. Acute effects of light on wellbeing and functioning (i.e., via non-SCN ipRGC projections), and 3. Effects of light on health via direct exposure to the skin. At a minimum, this overall structure should be outlined earlier, even if the authors do not want to include a full exploration of all possible mechanisms. My additional specific concerns are outlined below.
- In the abstract it would be good to explicitly outline the scope and intention of the review, rather than just presenting a summary of the authors findings. As it is I don’t think this aligns well with the structure (though I will suggest a restructuring) – the abstract should more clearly map out the topics to be covered. The authors do not mention the circadian clock here, which is central to the effects of light on health.
- The writing throughout is quite colloquial – and there are many statements which are unreferenced that clearly refer to specific works. These are too numerous to list, but the authors should closely review the work to ensure they are providing adequate supporting evidence. E.g, there are none in the visual perception section, very few in the section on the human body, etc.
- The authors state “a dysregulation of melatonin production can lead to alterations in the circadian rhythm, i.e. the sleep-wake cycle” – this is a problematic statement. Melatonin is one of a myriad of circadian rhythms, and is not the same thing as the “sleep-wake cycle”. These are related, but separable concepts. The suppression of melatonin is problematic on it’s own, AND may also mean that circadian rhythms are disrupted, and both of these things will have consequences for sleep. Circadian rhythms should never be referred to as singular, unless referred to one specific rhythm, e.g., the circadian rhythm in melatonin.
- The authors use the phrase “psychic functioning”, I think you mean psychological functioning, which is a more accurate/accepted term.
- The authors allude to but do not discuss the impacts of artificial lighting, this should be expanded on in the light and health section where they discuss lifestyle changes. A relevant reference (and many others which are cited here) is: Cain, S.W., McGlashan, E.M., Vidafar, P., Mustafovska, J., Curran, S.P.N., Wang, X., Mohamed, A., Kalavally, V., and Phillips, A.J.K. (2020). Evening home lighting adversely impacts the circadian system and sleep. Scientific Reports
- In the “light and mind” section, there are many unreferenced claims/statements, and the section should not start with “thus” as this is a linking work and this is a new section. There are mention of many studies of in-patients and light exposure which are not referenced.
- It is unclear why the outdoor and mental health section is separated from others – this section is quite limited and doesn’t seem to really focus on light, I would omit this or include relevant studies in other sections.
- The authors do not acknowledge anywhere the myriad of projections of ipRGCs to brain areas involved in the regulation of mood (e.g., the amygdala and habenula) which almost very likely underlie the positive effects of light on mood. It is implied instead that this occurs due to vitamin d synthesis – while this may contribute, it is a very incomplete explanation.
- There is an incomplete discussion of effects of serotonin on circadian rhythms, where perhaps a discussion of the data on serotonin and light may be more appropriate, some example papers which may be of interest are below:
McGlashan, E.M., Nandam, L.S., Vidafar, P., Mansfield, D.R., Rajaratnam, S.M.W., and Cain, S.W. (2018). The SSRI citalopram increases the sensitivity of the human circadian system to light in an acute dose. Psychopharmacology 235, 3201–3209.
Cuesta, M., Clesse, D., Pévet, P., and Challet, E. (2009). New light on the serotonergic paradox in the rat circadian system. J Neurochem 110, 231-243.
- In the “circadian system and melatonin” section, there is a typo, where it should read that the higher the ALAN intensity is, rather than highest
- In the Light Therapy section a rather oversimplified description of light therapy is given, where actually the implementation of the therapy is quite varied (e.g., it is not always delivered in the morning).
- The authors use a number of acronyms which are not always standard in the field, this is unnecessary and breaks up the writing – e.g., I suggest removing CD and LT. These can be spelled out.
Reviewer 2 Report
The authors provide a non-systematic review of how light can directly/indirectly affect human physiological and psychological health. Non-visual effects of optical radiation on human health and well-being is a relatively recent field, and one that is yet to be completely understood. Lack of clarity in the purpose of the manuscript, improper understanding and segregation of the various optical radiation components and failing to specify and convey the role of various lighting characteristics experienced throughout the day dampened my excitement for this review.
Some comments -
Line 23 - There is no "artificial" light. Throughout the manuscript, authors should consider using "electric" light if they are not comfortable with just "light".
Line 36 - Authors never specify the scope of the review - Is it limited to natural light (e.g. daylight, sunlight)? Is it about electric light? Is it about UV? or, is it about optical radiation in general?
Line 47 - Missing reference for "light" definition; There are more than one (CIE, IES)
Line 49 - This statement is inaccurate. There are several other properties of light - reflection, transmission, diffraction.
Line 52 - Missing references for neurophysiology mentioned. In the interest of keeping this review short, I would suggest the author to carefully go through the manuscript and add references to all key arguments (check for words such as evidence, study, it has been shown that, knowledge, etc.), as there are several instances of missing references. If you are citing a review paper, you should also cite the actual study from that review paper. (Lines 108-120 for instance)
Line 57 - This statement is inaccurate. Light, by itself, has no color.
Lines 67-70 and several other similar instances - This statement is inaccurate. Authors confound "light" with "UV" throughout the manuscript. Light is limited between wavelengths 380-780nm. UV is 100-400nm. Light does not help in synthesis of Vitamin D. Light can only be perceived through photopigments within the eyes in humans.
Line 72 - Light does not trigger synthesis of melatonin (and other hormones). Endogenous circadian rhythm of melatonin persists even in absence of light-dark cycle. It can, however, be phase entrained by tailored lighting interventions.
Lines 75-56. Authors confound circadian rhythm with sleep-wake cycle. Melatonin, cortisol, sleep-wake, etc. all exhibit circadian rhythmic behavior, regulated by the master pacemaker located in the hypothalamic SCN.
Line 78 - Misleading statement. There is not enough evidence to conclusively say "sunlight" can prevent diseases. Also, authors need to understand that the photoreceptors within the eye cannot tell the difference between a photon from sunlight or a photon from electric light.
Line 83 - "interesting results"? Good / bad?
Line 122-135 - Irrelevant - Not really about light (far-fetched at the best) . Authors should cite and discuss studies directly looking at outdoor light exposures and health outcomes.
Line 139 - Bit misleading as direct or indirect exposure is not the key point. Authors need to be more specific in regard to which lighting characteristics changed going from direct to indirect exposure? Was it the decrease in light level?
Line 141-142 - Irrelevant - Not really about light. Authors need to be mindful about deviating from the scope of this manuscript.
Line 148 - Why only "natural"? And, as I mentioned before, in humans, light (380-780nm) can only be perceived by the retinal photoreceptors and not indirectly.
Line 167 - IpRGCs are the primary conduits for circadian phototransduction, but they also receive and integrate signals from the classical photoreceptors (rods, cones)
Line 177-201 - Irrelevant - Not really about light. SCN is capable of oscillating endogenously, in absence of light-dark cycle and this review is not about SCN.
Line 202-217 - UV is not light. So again, does not align with the scope of the review.
Line 218-261 - Effect of light on circadian system not discussed at all. It is important to note that the human circadian system has a much higher threshold than the human visual system. In other words, a person can function visually, while being in circadian darkness. Further, in regard to the effect of light in melatonin, authors only mention that it can be suppressed by LAN, and then proceed on with a review of melatonin which does not align with the scope of the manuscript. Authors should have instead specified how amplitude of the nighttime circulating melatonin levels change as a function of different lighting characteristics (amount, spectrum, duration, timing).
Line 279 - Is it speculation? Missing reference? Authors switched to light pollution without explaining what it is. Also, shouldn't light pollution be very less at extreme (assuming remote) environments as compared to urban environment?
Line 280-290 - Here as well authors mention ailments associated with circadian disruption (not light). Inappropriate light exposure may or may not cause circadian disruption.
Line 298-299 - I would recommend the authors to review following article (along with all the cited works) - https://journals.sagepub.com/doi/abs/10.1177/1477153516682368 which clearly depict the significant advancements in regard to neuroanatomy and neurophysiology associated with circadian phototransduction.
Lines 311-312 - 10,000 lx for what application? Refrain from using Light Therapy generically. Different applications will undoubtedly require different light therapy interventions in regard to amount, spectrum, duration, timing. Population parameters, such as age, sex, individual optical characteristics and sensitivities, will further affect the efficacy of the lighting interventions. In other words, it is important to refrain from conveying to the reader than "one sock fits all" with Light Therapy. And by extension, the side effects will also change.
Line 358 - Changing architecture, by itself, is not relevant to light. Rather, authors can suggest how effectiveness of a lighting intervention can be increased by altering architecture within the space? For instance - positioning desks closer to the windows and facing them? Painting the room walls white to reflect more light within the space?
Reviewer 3 Report
The paper presents an overview of the effects of light in human health with more of a focus on mental health issues. In this respect it may have been useful to provide a more focused review. The paper seems to ignore a number of studies on the use of light in major depression / SAD for example. While it is difficult to review every study conducted the paper would benefit from the use of summary tables covering major themes. Furthermore the paper is some what uncritical in accepting findings from many of the studies discussed. In the treatment of depression / SAD for example while there do appear to be positive benefits a conundrum in such studies has been to find an acceptable 'placebo' against which compared light therapy.
There are a number of places where abbreviations are used without explanation e.g., PNEI abbreviation should not be used in the title of the paper.
Round 2
Reviewer 2 Report
Dear Authors,
Thank you for being diligent in addressing all previous concerns. I believe the quality and readability of the review has significantly increased. Just a minor comment -
Lines 115 (SSRI), 120 (DSM), please spell out the acronyms at first mention.
Best regards